# Challenges and Novel Opportunities of Radiation Therapy for Brain Metastases in Non-Small Cell Lung Cancer

**DOI:** 10.3390/cancers13092141

**Published:** 2021-04-29

**Authors:** Paola Anna Jablonska, Joaquim Bosch-Barrera, Diego Serrano, Manuel Valiente, Alfonso Calvo, Javier Aristu

**Affiliations:** 1Brain Metastases and CNS Oncology Radiation Medicine Program, Princess Margaret Cancer Center, Toronto, ON M5G 2M9, Canada; 2Department of Radiation Oncology, Clinica Universidad de Navarra, 31008 Pamplona, Spain; 3Department of Medical Oncology, Catalan Institute of Oncology, Doctor Josep Trueta University Hospital, 17007 Girona, Spain; jbosch@iconcologia.net; 4Girona Biomedical Research Institute (IDIBGI), Salt, 17190 Girona, Spain; 5Department of Medical Sciences, Medical School, University of Girona, 17071 Girona, Spain; 6IDISNA and Program in Solid Tumors, Center for Applied Medical Research (CIMA), University of Navarra, 31008 Pamplona, Spain; dserrano@unav.es (D.S.); acalvo@unav.es (A.C.); 7Department of Pathology, Anatomy and Physiology, School of Medicine, University of Navarra, 31008 Pamplona, Spain; 8Brain Metastasis Group, CNIO, 28029 Madrid, Spain; mvaliente@cnio.es; 9CIBERONC, ISCIII, 28029 Madrid, Spain; 10Department of Radiation Oncology and Protontherapy Unit, Clinica Universidad de Navarra, 28027 Madrid, Spain; jjaristu@unav.es

**Keywords:** brain metastases, non-small cell lung cancer, stereotactic radiosurgery, targeted therapies, radionecrosis, proton beam therapy, animal models

## Abstract

**Simple Summary:**

Lung cancer is the most common primary malignancy that tends to metastasize to the brain. Owing to improved survival of lung cancer patients, the prevalence of brain metastases is a matter of growing concern. Brain radiotherapy remains the mainstay in the management of metastatic CNS disease. However, new targeted therapies such as the tyrosine kinase or immune checkpoint inhibitors have demonstrated intracranial activity and promising tumor response rates. Here, we review the current and emerging therapeutical strategies for brain metastases from non-small cell lung cancer, both brain-directed and systemic, as well as the uncertainties that may arise from their combination.

**Abstract:**

Approximately 20% patients with non-small cell lung cancer (NSCLC) present with CNS spread at the time of diagnosis and 25–50% are found to have brain metastases (BMs) during the course of the disease. The improvement in the diagnostic tools and screening, as well as the use of new systemic therapies have contributed to a more precise diagnosis and prolonged survival of lung cancer patients with more time for BMs development. In the past, most of the systemic therapies failed intracranially because of the inability to effectively cross the blood brain barrier. Some of the new targeted therapies, especially the group of tyrosine kinase inhibitors (TKIs) have shown durable CNS response. However, the use of ionizing radiation remains vital in the management of metastatic brain disease. Although a decrease in CNS-related deaths has been achieved over the past decade, many challenges arise from the need of multiple and repeated brain radiation treatments, which carry along not insignificant risks and toxicity. The combination of stereotactic radiotherapy and systemic treatments in terms of effectiveness and adverse effects, such as radionecrosis, remains a subject of ongoing investigation. This review discusses the challenges of the use of radiation therapy in NSCLC BMs in view of different systemic treatments such as chemotherapy, TKIs and immunotherapy. It also outlines the future perspectives and strategies for personalized BMs management.

## 1. Introduction

Brain metastases (BMs) arise from the seeding of circulating tumor cells from a primary tumor into the brain microvasculature and cancer cell migration across the blood-brain barrier (BBB) [1]. The presence of BBB prevents many cytotoxic agents from entering into the central nervous system (CNS), thus creating a reservoir for the micrometastatic spread and contributing to the historic idea that the brain is a sanctuary for cancer cells [2]. 

The highest incidence of BMs is found in lung cancer (20–56%), followed by breast carcinoma (20–30%) and melanoma (5–10%) [3]. Approximately, 25–40% of patients with non-small cell lung cancer (NSCLC) develop BMs during the course of their disease [4]. Moreover, lung adenocarcinomas spread to the brain more frequently than squamous cell carcinomas [5]. BMs remain an important cause of morbidity and mortality in lung cancer patients. The incidence of BMs in NSCLC patients reported in past studies might have been underestimated, as a wider use of brain magnetic resonance imaging (MRI) in recent years has led to increased BMs detection. Another reason for that is the development of more effective systemic treatments that result in longer patient survivals and hence, in a larger population at risk for BMs development. 

Radiation therapy (RT) for CNS metastases has increasingly moved towards the use of more focused and accurate techniques, such as stereotactic radiosurgery (SRS) or hypofractionated stereotactic radiotherapy (hfSRT). SRS delivers a high, ablative dose of radiation with biologically effective doses (BED) greater than 100 Gy and a steep dose gradient to a precisely defined brain target, obtaining local control rates as high as 70–90% [6]. Upfront whole brain radiation therapy (WBRT), historically considered as first-choice therapy for multiple BMs, is now preferred in patients with extensive intracranial spread that is unlikely to be controlled by focal brain radiotherapy. Many studies have shown the advantage of using SRS for a limited number of BMs (1–3) in patients with good performance status and controlled primary disease, avoiding the adverse neurocognitive effects associated with WBRT [7,8,9]. Recent findings of a phase III trial presented at the ASTRO Annual Meeting provided supportive evidence for the use of SRS for patients with 4–15 BMs. Relatively to the WBRT, SRS was associated with a reduced risk of cognitive decline without compromising the overall survival (*NCT01592968*).

BMs diagnosis in stage IV NSCLC is an indicator of poor prognosis. However, the growing experience in this field shows that some of the NSCLC patients with favorable clinical and molecular factors achieve longer survival and maintain a durable control of CNS metastases [10]. Advances in the management of lung cancer have generated a new paradigm in the clinical practice [11]. Although the sequential or simultaneous combination of systemic and brain-directed therapies still remains under an early investigation, it appears a promising strategy for slowing down the intracranial spread. The use of SRS along with novel systemic agents that show a greater CNS penetration constitutes a new complementary and synergistic approach. At the same time, the potential toxic events such as the increased rate of radionecrosis (RN) are of concern, even though the evidence on that remains insufficient [12,13,14]. 

It is worth mentioning proton beam therapy as an evolving radiotherapy modality with the dosimetric and physical advantages of reducing or eliminating the dose to the surrounding tissue and organs in comparison to the conventional photon radiotherapy. Proton SRS was proved feasible in the treatment of BMs [15], but its role and selective utility are still to be determined. 

New strategies for the management of BMs in NSCLC patients are continuously arising and clinicians encounter significant uncertainties and challenges in this particularly heterogeneous cancer population. These might include:identification of patients who would benefit most from the combination of systemic and local brain-targeted radiation therapies;management of the potential toxicity when combining SRS with systemic therapies i.e., treatment of radionecrosis;limitation of the long-term sequelae of brain radiation in patients with stage IV NSCLC who achieve prolonged survival;risk of leptomeningeal or pachymeningeal failure with the increased use of focal brain radiation over WBRT;decision for a “watch and wait” approach of asymptomatic BMs when administering systemic drugs that exhibit intracranial activity.

This review focuses on discussing the systemic treatment strategies of NSCLC-BMs in combination with brain radiotherapy, stressing the opportunities and challenges of these therapies. 

## 2. Rationale for Combining Brain RT and Systemic Therapies and Putative Pitfalls

Systemic cancer therapies constitute the cornerstone of the treatment for disseminated tumors. However, reaching the brain tissue in significant therapeutic concentrations is hindered by the presence of the BBB, which strictly regulates the homeostasis of the CNS. Additionally, the microenvironment surrounding the tumor cells nesting within the brain confers an additional resistance that needs to be overcome [16]. The formation of BMs results in a disruption of the BBB architecture, deriving in a new structure called the blood–tumor barrier (BTB), considered to be more permeable than the BBB. However, the porosity and functionality of BTB remains heterogeneous and varies between different metastatic lesions and primary tumor types [17]. In spite of the enhanced BTB permeability, most systemic cancer drugs are unable to cross this barrier to reach the tumor in sufficient concentrations [18]. It is probably the principal reason why chemotherapy has not been effective in the treatment of BMs in NSCLC patients. Many chemotherapy drugs constitute large hydrophilic molecules that cannot traverse the CNS unless they are incorporated through a receptor-mediated endocytosis [1]. In addition, the multidrug resistant transporters in the BBB hinder further incorporation of systemic drugs into the CNS [19]. 

Ionizing radiation causes BBB disruption and potentially facilitates the incorporation of systemic agents into the brain. Doses of 20 to 30 Gy at 2 Gy per fraction have been shown to increase the BBB permeability [20]. An initial study published in 1990 by Qin et al. observed that the degree of destructive effect on the BBB in the normal irradiated tissue was directly proportional to the radiation dose [21]. Authors showed that the BBB in the tumor area was partially destroyed on an average of 22%, and that brain RT could increase this destruction to an average of 75%. Based on these findings and similar results published, the delivery of brain RT before initiation of the systemic treatment has been suggested in order to enhance drug penetration into the brain. More recent preclinical experiments suggest that extreme hypofractionation radiation schemes that use a single dose of 20 Gy per fraction are capable of increasing the permeability as early as 24 h upon treatment, but this effect could be delayed even up to 90 days [22]. However, there is still little understanding about the optimal timing of RT to reach the maximum BBB permeability as well as the time it takes to revert its effect. The impact of ionizing radiation on the BBB seems evident, whereas the adequate dose to achieve the desired outcome remains unclear. 

In contrast to chemotherapy, targeted therapies have shown high intracranial tumor response rates. The incidence of BMs in NSCLC patients with epidermal growth factor receptor (EGFR) mutations is higher than that observed in patients who are EGFR-wild type (70% vs. 38%, respectively) [23]. Response rates of BMs to EGFR tyrosine kinase inhibitors (TKIs) in patients harboring EGFR mutations are meaningful, with values ranging from 60% to 100% [18,24,25]. Nonetheless, BBB penetration for some of these agents, such as for the first-generation TKI gefinitib was proved low [26]. Erlotinib showed some improvement in CNS penetration, with up to 7% incorporation rates [27,28]. Second-generation EGFR-TKI afatinib has also demonstrated poor BBB penetrance [29]. More recent data confirm that third-generation EGFR-TKIs overcome the hurdles of resistance to first-/second-generation TKIs. Osimertinib was approved for second-line treatment of EGFR-mutant NSCLC following failure with earlier-generation EGFR-TKIs due to the acquired T790M mutation. This drug has at least 9-fold-increased BBB penetration compared to first- or second-generation TKIs [30,31]. AZD3759 represents a novel class of EGFR-TKI that cannot be expelled by multidrug resistant transporters. This compound was widely investigated both in preclinical and Phase I studies [32,33,34,35].

A high percentage of BMs is also found in patients with anaplastic lymphoma kinase (ALK) translocations. Approximately 35% of ALK-positive stage IV NSCLC is diagnosed with BMs at presentation. Moreover, up to 50–60% patients develop BMs later during the course of the disease [36]. Crizotinib has shown a significantly higher intracranial control rate compared to chemotherapy in patients with BMs at baseline [37]. Despite this, 34–50% of patients on treatment with crizotinib show CNS as the first and sole site of progression, and up to 70% of patients free of BMs at baseline develop CNS metastases later on [38,39]. The low intracranial activity of Crizotinib compared to other organs can be again explained by poor BBB penetration [39,40]. Second generation ALK-TKIs alectinib and ceritinib showed improved CNS incorporation (15% and 63–94%, respectively). The third-generation drug lorlatinib has demonstrated a 75% incorporation rate [41].

In addition to the BBB penetration, experimental models have proven that the combination between RT and EGFR-TKIs or ALK-TKIs has a synergistic antitumor effect due to intrinsic mechanisms in cancer cells, which is another reason that supports the rationale of combining RT with TKIs. Osimertinib combined with RT in the EGFRT790M mutant NCI-H1975 NSCLC model revealed enhanced antitumor activity as compared to single therapies [42]. The underlying mechanism was found to be the inhibition of proliferation following irradiation, as well as delayed DNA damage repair [43]. Mechanistically, AZD3759 radiosensitized NSCLC cells had inhibited both the non-homologous end joining (NHEJ) and homologous recombination (HR) DNA double-strand breaks (DSBs) repair pathway, thus suppressing DNA damage repair. Using preclinical models of ALK-positive NSCLC (H3122) and ALK-negative NSCLC (A549 and LLC), Dai et al. showed synergistic effects in combination with RT only in the ALK-positive model, in vitro and in vivo [44].

With respect to the possible advantages of combining RT with immunotherapy, both experimental and clinical evidence show positive results. RT is known to activate the anticancer immune response by several mechanisms that involve both the tumor cells and the tumor microenvironment [45]. Many preclinical studies have demonstrated immunomodulatory effects of RT when applied to primary malignancies, including the upregulation of programmed death-ligand-1 (PD-L1) and inflammatory cytokines, release of tumor-associated antigens and increase in T cell infiltration. When combined with anti-PD-1, RT improves the antitumor effect in both the irradiated area and even in non-irradiated tumors through the abscopal effect [46]. 

Recent publications have given more insight into the CNS immunology, suggesting that the brain is not entirely immunologically isolated. There is a lymphatic drainage system in the meninges that allows for the CNS antigens to reach the cervical lymph nodes, while activated T-cells can cross the BBB and return to the systemic circulation [47]. Although T cell infiltration and activation within the CNS seem to be limited, the use of combination therapies, whichmay increase antigen presentation of T cells could be a promising strategy. An anecdotic case described a patient diagnosed with NSCLC and a single brain metastasis who showed thoracic response following SRS treatment for the brain lesion [48]. The use of monoclonal antibodies targeting immune checkpoint inhibitors (ICIs) in NSCLC patients with BMs shows encouraging results [49]. These molecules have traditionally been consideredincapable of crossing the BBB. Nonetheless, the mechanism of action of ICIs in BMs is still not fully understood and may be highly related to the immune trafficking of effector lymphocytes from other extracranial locations.

## 3. Combination Strategies Using Radiotherapy 

### 3.1. Chemotherapy

As previously mentioned, the majority of the chemotherapy agents have a restricted intracranial activity and thus a limited effect on NSCLC-BMs. The alkylating agent temozolomide (TMZ) is clearly active against high-grade glioma and is capable of reaching elevated concentrations in the CNS both before and after RT [50]. Unfortunately, it is not as effective in NSCLC patients, withs the response rates falling below 10% [51]. Some studies explored the use of TMZ with concomitant WBRT, with preliminary results suggesting that this combination might improve the BMs local control rate and PFS [52,53]. 

Other well-known antineoplastic drugs such as cisplatin have been proven to cross the BBB in NSCLC patients, achieving response rates of 15–30% [54]. The antifolate pemetrexed in combination with cisplatin or carboplatin and the recently approved ICI pembrolizumab constitute the first-line treatment in advanced NSCLC [55,56,57]. Pemetrexed combined with cisplatin is also an effective frontline chemotherapy in poor-prognosis non-squamous NSCLC patients, including those diagnosed with inoperable or SRS-ineligible BMs [58]. Its safety and efficacy in combination with pembrolizumab for NSCLC patients with poorer performance status has also been investigated [59]. In a retrospective study of thirty NSCLC patients without previous brain radiation, first-line pemetrexed and cisplatin/carboplatin showed intracranial response of 33.3% after two cycles, with an overall partial response of 23.3% and a median time to brain progression of 6.0 months [60]. BBB permeability and distribution of pemetrexed within the brain are limited: cerebrospinal fluid (CSF) concentration less than 5% of the concentrationfound in plasma was determined after the intravenous administrationin cancer patients [61,62]. This suggests that effective drug combinations between pemetrexed and cisplatin or the anti-PD1/anti-PDL1 agents can result in intracranial responses in spite of the expected low BBB penetration rates [18]. A Phase II trial evaluating the efficacy and safety of pemetrexed-cisplatin plus concurrent WBRT in patients with non-squamous NSCLC and BMs (n = 42) observed an intracranial response of 68.3% and extracerebral overall response rate of 34.1%, with a median BMs PFS of 10.6 months and a median OS of 12.6 months [63]. As chemotherapy remains an essential treatment in NSCLC patients, its use in combination with both RT and ICIs for the treatment of BMs needs more investigation. 

### 3.2. Targeted Therapies

The use of EGFR-TKIs alone is associated with high intracranial response rates, while the benefit of adding brain radiation is being investigated [64]. Most of the initial studies that addressed this issue were conducted on WBRT and erlotinib. A study by Zhuang et al. included NSCLC patients with multiple BMs treated with WBRT alone or in combination with erlotinib (n = 54). The findings showed a significant benefit for the combination treatment in terms of overall response rate (ORR, 54.89% vs. 95.65%, *p* = 0.001), median PFS (5.2 vs. 10.2 months, *p* = 0.003) and median OS (8.9 vs. 10.7 months, *p* = 0.020) [65]. In contrast, the TACTIC trial evaluated the effect of WBRT alone or in combination with erlotinib in unselected NSCLC patients with BMs (n = 80). PFS and OS were similar for both groups, suggesting no benefit for the combination arm [66]. However, only 2.9% of patients were proved to have activating EGFR mutations.

A more recent study by Cheng et al. retrospectively analyzed 78 patients with EGFR-mutant lung adenocarcinoma who developed BMs and received either a combination of EGFR-TKIs and RT (WBRT or SRS) or EGFR-TKI alone. The median time to intracranial progression was longer in the combination group than that of the TKI group alone (21.5 vs. 15 months; *p* = 0.036) [67]. However, PFS and OS did not differ. Another retrospective study used gefitinib in combination with WBRT in 90 patients, showing superior PFS (10.6 vs. 6.57 months, *p* < 0.001) and OS (23.40 vs. 14.93 months, *p* = 0.02) when compared to gefitinib alone. ORR was also significantly better in the combination arm (64.4% vs. 27.6%, *p* < 0.001) [68]. Zhen et al. conducted a meta-analysis including five studies comparing the use of WBRT plus TKI vs. WBRT alone and five other studies that looked at WBRT plus TKI vs. TKI alone in unselected NSCLC patients with BMs [69]. EGFR-TKI alone provided similar OS and extracranial PFS when compared to WBRT plus EGFR-TKI, but superior intracranial PFS. However, only two out of the ten studies reported the results of intracranial PFS. Of note is the Phase III RTOG 0320 trial that tested TMZ or erlotinib in combination with WBRT plus SRS in unselected patients [70]. Results showed that the addition of either TMZ or erlotinib did not improve survival and could be associated with a higher incidence of grade ≥3 toxicity, but these conclusions should be interpreted carefully due to the lack of statistical power. Zhao et al. compared the therapeutic effect of WBRT on advanced NSCLC between EGFR TKI-naïve and TKI-resistant patients [71]. The authors found no differences in intracranial PFS and OS between the two. Interestingly, a subgroup of patients with a lung-molecular graded prognostic assessment index (Lung-molGPA) of 2.5–4, indicating a better prognosis, had a better intracranial PFS and OS if they were TKI-naïve.

Current clinical trials are prospectively evaluating the effects of combining SRS and EGFR-TKIs (with or without WBRT) in patients with known EGFR status (Table 1). Although the effectiveness of concurrent RT and EGFR-TKIs in BMs from NSCLC patients remains inconclusive, results from these trials will reveal whether these strategies are clinically plausible. The development of EGFR targeting agents designed to penetrate the BBB (such as AZD3759) may also change the management of patients with BMs in combination with RT [43]. 

With regard to ALK rearrangement, the PROFILE trials have shown that the first-generation TKI crizotinib obtained high rates of extracranial ORR in ALK-positive stage IV NSCLC patients (59.8–74%), but the effect on BMs was much more modest [72]. The pivotal trial PROFILE 1014 obtained a survival probability at 4 years of 56.6% (95% CI, 48.3–64.1%) with crizotinib and 49.1% (95% CI, 40.5–57.1%) with chemotherapy. These results are in keeping with the findings of a cohort study reported in 2016, where 90 ALK-positive NSCLC patients with BMs at baseline showed a median OS of 49.5 months (95% CI 29.0–not reached) despite a median intracranial PFS of 11.9 months (95% CI 10.1–18.2) [38]. Yoshida et al. reported ORRs of 66% and 20% in extracranial and intracranial lesions, respectively [73]. Combination of crizotinib and brain RT significantly improved ORR in ALK-positive NSCLC patients (from 18% to 33%) and also the median time to tumor progression (from 7 to 13.2 months) [74]. 

Second-generation ALK-TKIs such as ceritinib, alectinib and brigatinib, as well as the third-generation ALK-TKI lorlatinib have shown greater efficacy in BMs control. Alectinib was more active than crizotinib in first line for advanced ALK-positive NSCLC patients, especially in patients with BMs at diagnosis [75]. In addition, alectinib reduced the cumulative risk of developing BMs. Despite this, there was an increased risk of 12-month cumulative incidence rate of CNS progression to first line alectinib: from 8.6 to 20.5%, among those who had received previously brain RT and those who had not [75]. The efficacy of combining RT with next-generation ALK-TKIs has not been studied in depth. The ASCEND trials have explored the effect of ceritinib in combination with RT in BMs, showing no difference between the study groups [76,77]. However, these were retrospective studies with small sample size. 

Additional targeted therapies such as VEGF (bevacizumab, endostar), as well as PARP (veliparib) and mTOR (everolimus) inhibitors are being tested in combination with WBRT [78]. A recent systematic review by Peravali et al. showed that the combination of EGFR-TKI plus VEGF inhibitor provided a significant improvement in PFS, but not in OS or ORR, when compared to EGFR-TKI alone in stage IV NSCLC [79]. However, data on BMs and brain RT were not reported. For all targeted therapies, concerns regarding the optimal timing for the combination, type of brain RT (whether WBRT or SRS) and toxicity still need to be investigated in future studies. There are data that suggest a possible benefit from simultaneous administration of brain radiation and TKIs in terms of improved intracranial control. However, various studies report results including both WBRT and SRS techniques, where the radiation-induced cell death, effects on BBB permeability and radiobiology mechanisms differ. 

### 3.3. Immunotherapy

Immunotherapy alone is showing encouraging results in patients with driver-negative, PD-L1-positive advanced NSCLC with limited BMs. CNS responses of around 30% have been reported with single ICIs such as pembrolizumab, which is similar to the responses achieved extracranially [80]. There are divided data on the effect of combining immunotherapy and brain RT in NSCLC patients. Most studies are retrospective in nature. Singh et al. reviewed 85 NSCLC patients with BMs treated with SRS and anti-PD1 therapy. Unlike in melanoma patients, there was no significant benefit of combining SRS with an anti-PD1 agent in terms of survival or radiological response. Lesions greater than 500 mm^3^ were found to show a faster volumetric response, which can be particularly beneficial for BMs in neurologically compromised locations or those causing significant mass effect. On the contrary, other clinical data suggested improvement in OS with concurrent ICIs and RT when immunotherapy was started at least 30 days prior to RT and continued throughout the radiation treatment [81]. One of the largest NSCLC series that was treated with anti-PD1/anti-PD-L1 therapy and different modalities of radiation (SRS, WBRT, partial brain radiation) proved this combination to be safe and effective [82]. Other retrospective studies suggest that rates of RN are slightly higher with immunotherapy combinations than without, but these data are conflicting [13,14,83,84]. A meta-analysis from 2019 reported a 0% to 21% risk of RN in patients with BMs treated with SRS and ICIs [85]. One of the caveats is that most of the studies involve different primary tumor histologies (NSCLC, renal cell carcinoma, breast and colorectal cancer), which use different systemic agents. Furthermore, many studies report on the overall treatment toxicity, and not the incidence of radionecrosis specifically.

Research findings on the clinical benefit of SRS or WBRT in combination with immunotherapy remain divided. In the past few years, the management of brain disease in patients with lung cancer has evolved, with a trend towards more focal radiation treatments such as SRS, surgical resection followed by SRS or multiple SRS alone instead of WBRT, which is associated with a short-term cognitive decline. However, this is not a universal practice and drawing solid conclusions from the current data remains challenging. Several ongoing trials will be able to provide more evidence on the use of immunotherapy with brain radiation techniques when managing CNS metastases (*NCT02085070, NCT02681549, NCT02978404, NCT03366376, NCT02858869, NCT03325166, NCT02696993, NCT02831959, NCT01454102, NCT02320058, NCT02374242* and *NCT02621515*).

## 4. Oligometastatic CNS Disease and “Watch and Wait” of Asymptomatic BMs

The definition of oligometastasis refers to a limited number of metastatic disease burden that can be safely treated with metastasis-directed focal therapies. There are different subtypes of oligometastatic disease (OMD), which might be relevant when treating BMs in EGFR/ALK+ NSCLC patients. Recently, ESTRO/EORTC and ASTRO have developed consensus documents to standardize the different terms used in OMD [86,87]. According to the time of diagnosis and systemic treatment, first development of BMs within 6 months of primary cancer is classified as synchronous OMD. In contrast, patients on active systemic treatment, with no prior OMD and BMs diagnosis over 6 months after primary cancer, are defined as metachronous oligoprogressive. Finally, the same case scenario but in absence of active systemic therapy is denominated as metachronous oligorecurrence. A recent work by a Canadian group has highlighted the importance of distinguishing between the different subtypes of OMD [88]. Each entity might entail different prognosis and thus require adopting a different treatment strategy. There is evidence suggesting that oligoprogressive disease is a clinically distinct state, resulting in worse outcomes [87].

Synchronous oligometastatic NSCLC represents a distinct category of patients that is known to achieve better survival if all the sites of macroscopic disease are treated radically. First prospective data from 39 patients were published in 2012 by De Ruysscher et al., claiming that the identification of this favorable subgroup before therapy was essential [89]. A subsequent study from Massachusetts compared 186 NSCLC patients presenting with OMD versus 539 patients with extensive NSCLC disease (more than 5 distant metastatic lesions) [90]. The aggressive local therapy of primary disease (defined as surgical resection and/or definitive radiation) was associated with prolonged survival (HR, 0.65, *p* = 0.043) in the OMD group. Patients with BMs represented 29% of the oligometastatic cohort. Good performance status, nonsquamous histology, and limited nodal disease were predictive factors for improved survival. Many other studies have focused on synchronous brain-only oligometastasis or included mainly brain oligometastatic patients in their analysis, confirming these results [91,92,93].

As the standard therapeutic approach for oligometastatic NSCLC patients remains undefined, a rigorous discussion on individualized therapeutic management is fundamental. A recent work by an Italian group suggested the urge of adequate selection criteria for this subgroup in order to discuss the potential benefit of systemic and radical loco-regional treatments [94]. This study analyzed a total of 281 synchronous oligometastatic NSCLC patients from different centers, who received radical surgical treatment of the primary tumor with or without neoadjuvant/adjuvant therapy, and radical treatment of all metastatic sites. Brain was the most common site of oligometastasis (50.9%). The 2- and 5-year overall survival was 60.4% and 36.1%, respectively. The authors concluded that young age, single metastasis and pN0 disease were factors associated with better prognosis.

Of note is the first multicenter randomized Phase II trial published by Gomez et al. in 2016, which compared aggressive local therapy to standard-of-care maintenance treatment or observation in oligometastatic NSCLC patients, that did not progress after first-line systemic treatment [95]. This study was terminated early due to substantial efficacy improvement in the local consolidative therapy group. Thirteen (26.5%) out of forty-nine randomized patients had metastasis to the brain. The median progression-free survival in the local consolidative therapy group was 11.9 months versus 3.9 months in the maintenance treatment group (HR 0.35, *p* = 0.0054). While awaiting results of ongoing Phase III trials, the decision on primary tumor management should be made on an individual basis, with support of the available evidence. In addition to a widely implemented radical management of BMs, primary local tumor treatment should be strongly considered after identification of this particular subgroup of patients. A multidisciplinary consensus statement on definition and staging of synchronous oligometastatic NSCLC has been formulated and should be used for this purpose [96].

CNS oligoprogression is not uncommon, especially in EGFR/ALK+ NSCLC. To our knowledge, there are limited data on NSCLC patients with metachronous BMs harboring driver mutations. Patients who receive first-line TKIs are at higher risk of developing new BMs due to acquired mutations and flare phenomenon [97]. Real-world data from a Korean study of 22,458 NSCLC patients showed that after 10 months of first-line systemic treatment, patients receiving targeted therapies had a significantly higher cumulative incidence of BMs [98]. To the best of our knowledge, there are no data to support the therapeutic management of repeat oligoprogression or repeat oligopersistance when on a TKI agent only. The right timing of radiation remains a matter of debate. Decision for a “watch and wait” approach of stable and limited BMs might be challenging, given the existing risk of a rapid CNS dissemination. On the other hand, as described in previous sections, there is sufficient evidence that third-generation TKIs such as osimertinib achieve durable CNS response and allow to hold up potential brain radiotherapy. Close MRI surveillance plays a crucial role in detecting early progression and guiding the next steps. One of the pitfalls of delaying potential focal radiotherapy is that patients initially considered candidates for ablative local treatment can rapidly develop widespread CNS disease, for which SRS or fSRT would no longer be appropriate. A multi-institutional study by Magnuson et al. showed that upfront SRS in EGFR+ NSCLC yields better overall survival [99]. However, this should be contemplated in view of lacking strategies to identify oligometastatic CNS patients, who develop more aggressive and rapidly progressive BMs with a potential for leptomeningeal failure. In addition, clinicians might be reluctant to observe small lesions located in eloquent brain areas such as motor strip or brainstem, as future progression of these can lead to significant neurological symptoms and functional decline. A treatment algorithm for the management of de novo and progressive BMs has been suggested (Figure 1). Additionally, factors required for an integrated evaluation on BMs treatment decision and most common clinical scenarios are represented in Figure 2 and Figure 3. 

## 5. The Challenge of Recurrence, Radioresistance and Radionecrosis 

In addition to the challenges of combining RT and systemic drugs, radiation treatments may encounter several other problems related to recurrence, radioresistance and radiation-related toxicity. As cancer patients with BMs achieve longer survivals, they might require repeated courses of brain radiation in order delay the intracranial progression for as long as possible. Reirradiation of targets that have failed locally can be challenging, especially if they are located near to other vulnerable structures that have already received high doses of radiation previously. Moreover, re-exposing healthy brain tissue to subsequent RT inevitably increases the risk of necrosis [100]. Symptomatic and asymptomatic RN have been reported approximately in 10% and 30% of patients, respectively, but the data vary. RN is a radiation-related effect that can be seen in 3 to12 months after delivering the high dose radiation, or even years after. BMs of ALK-positive patients are especially prone to developing RN (HR 6.36, *p* < 0.001) [101], but interestingly, the use of ALK inhibitors does not seem to be associated with a higher incidence of RN [101]. Similarly, combined cytotoxic chemotherapy does not increase the risk of SRS toxicity. On the other hand, other data suggest an increased risk of post-SRS necrosis with concurrent VEGFR TKIs and EGFR TKIs [102], as well as with immunotherapy, although not all studies are positive [13,14]. 

The etiopathology of RN is not fully understood, but direct and indirect effects of radiation on vascular, neuronal and immune cells seem to play a key role in its development. Ionizing radiation causes sphingolmyelinase-dependent apoptosis and production of reactive oxygen species (ROS) in endothelial cells, leading to increased permeability and edema. This generates a hypoxic condition that induces VEGF secretion by both endothelial cells and reactive astrocytes. Radiation also damages oligodendrocytes, causing demyelination. As a result of this damage, there is a local infiltration of immune cells that secrete a variety of pro-inflammatory cytokines, such as TNF-alpha, IL-6 and IL-1a. All of these events result in a massive necrotic cell death and extra-cellular edema, leading to non-specific symptoms such as seizures, headaches, nausea and vomiting, ataxia or focal neurological deficits depending on the affected brain area [103].

There is no ideal diagnostic tool to differentiate RN from tumor recurrence, as both of these entities display similar radiological characteristics on conventional imaging used for follow-up of patients with BMs. Functional imaging techniques such as perfusion or diffusion MRI and PET with different molecular markers can provide useful information to help in this differential diagnosis [104]. Another option is a surgical resection, which allows for tissue diagnosis and symptom alleviation by reducing the associated perilesional edema. However, it might be problematic in patients who have undergone previous craniotomies, of poor performance status and comorbidities. Patients with RN are initially managed with corticosteroids. Other options include hyperbaric oxygen therapy or the VEGF inhibitor bevacizumab [105]. A Phase II clinical trial is currently investigating the combination of corticosteroids and bevacizumab for the treatment of RN (*NCT02490878*). In addition, laser interstitial thermal therapy (LITT) constitutes a minimally invasive technique that uses ablation of intracranial lesions with stereotactic guidance. Its role in reducing RN has also been explored in the past [106]. 

SRS provides 1-year local control ofat least 75% for limited CNS disease in NSCLC patients [107]. There are no standard recommendations for the management of local failure, but the options include surgery, WBRT, repeat SRS or change in systemic treatment depending on the clinical context of the patient. SRS can also be delivered in combination with WBRT [108]. According to the published data, local control rates of the combination are higher when compared to the administration of SRS alone [109]. This may be related to the effect of fractionation, which would promote radiation resistance mechanisms, such as hypoxia.

Unfortunately, the most frequent clinical scenario is that of progressive brain disease, where patients develop new BMs outside of the previously treated field. Up to 60–80% of primary BMs present with failure out of the initially radiated area [8,109,110,111,112]. The management of recurrent, not disseminated BMs is not well-established either. In the past, WBRT has been the standard for salvage brain treatment with conflicting results [113]. Specific data on local and distant control rates after salvage WBRT are scarce, which can be possibly explained by the short survival of these patients [114]. More recently, a small number of trials were published, reporting on the use of SRS for re-irradiation of BMs after previous WBRT [115,116,117]. There are data on local control rates as high as 90% when using SRS for relapses after WBRT or repeated in-field SRS [118]. Still, the salvage approach in patients with limited brain progression needs further assessment and stronger recommendations on this common clinical situation.

## 6. Proton Beam Therapy for the Treatment of NSCLC Brain Metastases

The use of proton therapy is increasing in the management of cancer, with a rising number of registered proton centers worldwide. According to the last data published by the Particle Therapy Cooperative Group (PTCG) in September 2020, a total of 110 operating particle facilities were registered, 12 of which correspond to carbon ions units and 99 are proton therapy facilities. In addition, at least 64 particle beam centers are currently under construction or at a planning stage [119]. Proton-based radiation treatment offers the advantage over a standard photon therapy of minimizing the radiation dose in tumor-adjacent normal structures. Protons are particles with mass and charge and a physical property of increasing the ionization when crossing a tissue. Deposition of this energy rapidly reaches a maximum, with a very steep decline of the radiation dose behind the target volume, resulting in the characteristic Bragg peak [120]. In 2017, the American Society of Radiation Oncology published two updated categories with the insurance coverage recommendations for proton therapy (Table 2). 

Proton therapy is becoming an attractive therapeutic option for patients with lung cancer who require RT. It provides the possibility of reducing the radiation exposure to the organs at risk, especially to the heart, in NSCLC patients treated with radical or adjuvant thoracic radiation [121]. In the same way, protons allow to explore the dose escalation effects in terms of locoregional control and OS in patients treated with photon-based techniques without increasing the low radiation dose to the heart and lung and reducing the effect of lymphopenia, which has been shown to be an independent variable associated with poorer survival [122]. Recent publications have demonstrated the safety and effectiveness of proton treatment modality in patients with locally advanced NSCLC [123,124,125], which has led to the design of new prospective cooperative group studies analyzing proton therapy for this disease.

Proton therapy has the potential to reduce the long-term brain adverse effects such as RN, observed in patients treated with photon-based SRS. There are scarce data in the literature regarding the treatment of BMs with proton-SRS. Recently, a first series of NSCLC patients with single or multiple BMs treated with proton-SRS was published [15]. This retrospective study from the Massachusetts General Hospital (Boston, MA, USA) included 815 metastases from 370 patients and concluded that the procedure was well-tolerated. The findings in terms of local control were comparable with those of photon-SRS. It is still unclear which patients can benefit from brain integral dose reduction and whether or not radiation-related effects such as RN can be avoided. Therefore, searching for predictive biomarkers of late toxicity may shed light on the potential of proton-SRS treatment in reducing the incidence of RN. On the other hand, the therapeutic radiobiological mechanisms of protons on BMs and the effect on normal cells of the nervous and immune systems are not yet well understood. Considering all this, there is a need for well-designed trials to assess the clinical efficacy of proton-SRS in NSCLC patients with BMs, alone or in combination with other therapeutic strategies, as well as identification of predictive biomarkers of tumor response, brain metastasis control and toxicity. 

## 7. Animal Models to Test Novel Therapeutic Approaches

The use of animal models of NSCLC metastases to the brain could greatly help to understand the molecular biology underlying this specific tropism and serve as a preclinical platform to test novel therapeutic approaches to guide clinical trials. A comprehensive review on available models of BMs was recently published [126]. In contrast to other cancer types with high incidence of BMs, where numerous organotropic cell line-based models are available, such as breast and melanoma, only a few NSCLC models have been described. Moreover, only two of these models use murine cell lines that can be inoculated in syngeneic immunocompetent mice to assess immunotherapy. 

The majority of NSCLC-BMs models use human cell lines injected into immunodeficient mice and different inoculation routes: intracranial, intracardiac, intracarotid, intravenous or orthotopic in the lung. In these models, reporter systems such as luciferase expression are commonly used to detect the presence of BMs in vivo. Each of these methods has advantages and disadvantages in terms of similarity with the natural BMs development process and for therapy testing. In 2009, Nguyen et al. described two models of NSCLC-BMs, which usedintracardiac injection of human adenocarcinoma cells: PC-9-BrM3 and H2030-BrM3 [127]. These cell clones were generated by isolation of BMs cells after three rounds of sequential intracardiac inoculation.

Although several systemic therapies have been tested in these models, no studies on RT have been reported. Similar to the human NSCLC cells with brain tropism, clones from the murine Lewis lung carcinoma cell line (LLC-BrMet) highly metastatic to the brain have been generated. These cells can be injected in the carotid of C57/Bl6 immunocompetent mice to generate BMs [128] but other routes of administration have also been successful [129]. The LLC-BMs model has been used to study BBB permeability and the effect of RT [130]. BBB permeability was assessed at various stages of tumor development and the authors concluded that, although BBB integrity is altered at late stages, it remains functional and limiting to drug permeability during most steps of tumor growth. Ewend et al. reported that local delivery of chemotherapy using biodegradable polymers and concurrent RT prolonged mice survival in the LCC-BMs model [131]. A study where Gamma Knife radiation was applied to LLC-BMs together with a nanoparticle-mediated radiosensitizer has shown a potent antitumor response [132]. Recently, an experimental combination between the antiangiogenic drug endostar and stereotactic radiotherapy using intracarotid injection of LLC cells into female C57/Bl6 mice showed significant reduction of BMs in the combination group compared to single therapies. Endostar normalized tumor blood vessels and increased the anti-tumor immune-related cell infiltration following RT treatment [133]. Our group is actively working on establishing new syngeneic animal models of BMs where novel combination therapies can be tested. 

## 8. Conclusions and Future Perspectives

Although BMs in NSCLC patients remain an important cause of morbidity and mortality, the therapeutic management of CNS spread in this cancer population has drastically changed over the recent years. With increased survival, more NSCLC patients develop intracranial metastases during their cancer journey. Some of the novel systemic treatments for patients with specific gene mutations (i.e., EGFR/ALK mutations and PDL1 expression) show promising results, but further studies are needed to continue improving the BMs control in stage IV NSCLC. Moreover, the wider use of photon-SRS and development of other RT modalities such as proton-SRS, are expected to positively impact the management of brain disease, with less neurological deaths, toxicity and cognitive decline. The combination strategies using RT and systemic agents are currently being investigated as a new approach to manage BMs with the goal of enhancing efficacy without increasing toxicity. In this sense, the design of EGFR-TKIs and ALK-TKIs with higher BBB penetration may be critical for the success of such combinations. In the era of immunotherapy, clinical trials will determine whether ICIs in combination with RT will be synergistic for patients, although increased toxicity may have to be taken into consideration. One key aspect is to search for biomarkers to identify patients who might benefit most from the combination of local and systemic treatments. Additionally, based on patient-specific factors, this could help to predict the individual risk of toxicities such as the development radionecrosis. Both clinical trials addressing these issues and preclinical assays may help to improve and take the management of BMs in NSCLC patients to a higher level of efficacy.

## Figures and Tables

**Figure 1 cancers-13-02141-f001:**
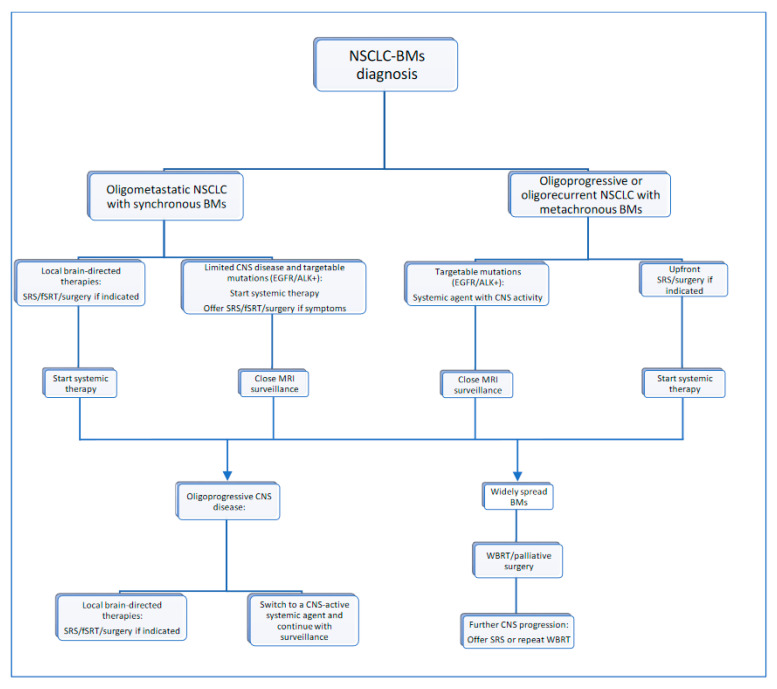
Proposed treatment algorithm for NSCLC-BM.

**Figure 2 cancers-13-02141-f002:**
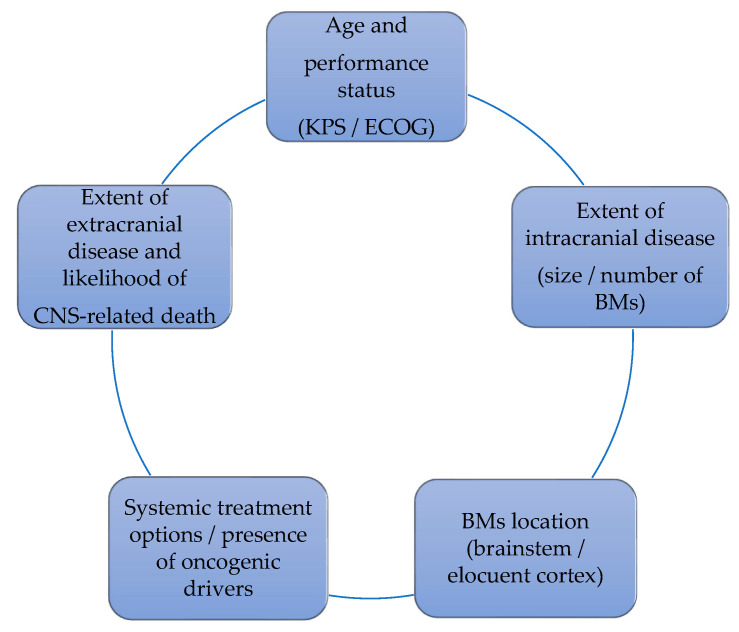
Factors that influence the decision on BMs management.

**Figure 3 cancers-13-02141-f003:**
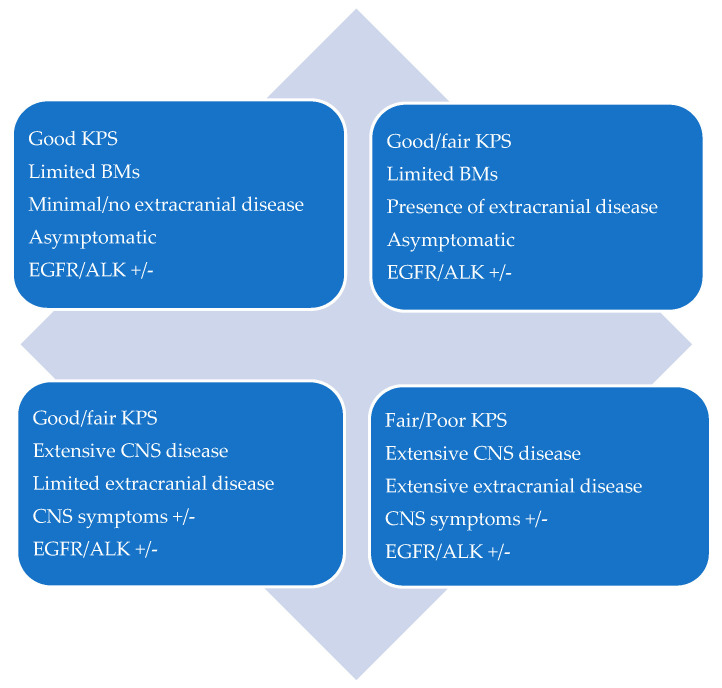
Patient scenarios.

**Table 1 cancers-13-02141-t001:** Ongoing combination trials with SRS and EGFR-TKI in EGFR+ NSCLC patients.

Trial Number	Phase	Key Eligibility	Intervention	Status
NCT03535363	I	EGFR-mutated NSCLC with 1-10 BMs	Osimertinib + SRS	Recruiting
NCT03497767	II	EGFR-mutated NSCLC with BMs diagnosed de novo or developed while on first-line EGFR-TKI	Osimertinib alone vs. upfront SRS + osimertinib	Recruiting
NCT02726568	II	EGFR-mutated NSCLC with BMs	Icotinib + SRS when intracranial progression	Recruiting
NCT03769103	II	Metastatic EGFR-mutated NSCLC with BMs	Osimertinib alone vs. osimertinib + SRS	Recruiting
NCT04193007	II	Asymptomatic NSCLC BMs with Gene-Sensitive Mutation	Molecular therapy alone (first-generation EGFR-TKI or ALK inhibitor) vs. molecular targeted therapy + brain radiotherapy (SRS or SMART-brain)	Not yet recruiting
NCT01573702	II	EGFR-mutant NSCLC patients who progressed on prior EGFR-TKI therapy	SRS or another local ablation followed by Erlotinib	Active, not recruiting

BMs: brain metastases; SRS: stereotactic radiosurgery; EGFR: epidermal growth factor receptor; NSCLC: non-small cell lung cancer; TKI: tyrosine kinase inhibitor; SMART: simultaneous modulated accelerated radiotherapy.

**Table 2 cancers-13-02141-t002:** ASTRO updates on insurance coverage recommendations for proton therapy.

Group 1 Indications *	Group 2 Indications **
Malignant and benign primary CNS tumors * Advanced (e.g., T4) and/or unresectable head and neck cancers * Tumors of the paranasal sinuses and other accessory sinuses * Nonmetastatic retroperitoneal sarcomas * Reirradiation cases where cumulative critical structure dose would exceed tolerance dose * Hepatocellular cancer (no longer required to be treated in a hypofractionated regimen *) Ocular tumors, including intraocular melanomas Tumors that approach or are located at the base of skull, including but not limited to chordoma and chondrosarcomas Primary or metastatic tumors of the spine where the spinal cord tolerance may be exceeded with conventional treatment/where the spinal cord has previously been irradiated Primary or benign solid tumors in children treated with curative intent and occasional palliative treatment of childhood tumors where one of the criteria noted above apply Patients with genetic syndromes to minimize the total volume of radiation, such as but not limited to NF-1 and retinoblastoma patients	Non-T4 and resectable head and neck cancer Prostate cancer Breast cancer Thoracic malignancies Abdominal malignancies, including nonmetastatic primary pancreatic, biliary and adrenal cancers Pelvic malignancies, including nonmetastatic rectal, anal, bladder and cervical cancers

* Clinical scenarios that frequently support the use of proton therapy based on medical necessity and published clinical data were updated with five additions and one modification. ** Coverage with Evidence Development (CED). These indications also represent the disease sites for which evidence is accumulating and may support future Group 1.

## Data Availability

Not applicable.

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
