# Peer review of "Challenges and Novel Opportunities of Radiation Therapy for Brain Metastases in Non-Small Cell Lung Cancer"

_cancers, 2021, doi:10.3390/cancers13092141_

Round 1

Reviewer 1 Report

Thank you for your valuable work.

  • Table 1 should be justified
  • I would have expected a discussion on the potential radiotherapy treatment of inactive brain metastases

Author Response

Response to Reviewer 1

Comment: I would have expected a discussion on the potential radiotherapy treatment of inactive brain metastases

Answer: Thank you for reviewing our work and your contribution to improving the quality of the manuscript. Table 1 has been justified, as suggested.

The potential radiotherapy treatment for persistent non-progressive brain metastases is indeed a matter of debate and the authors wanted to thank you for making this valid point. The discussion on this topic has been included in a newly created section on oligometastatic CNS disease, see page 9, last paragraph. We have also included 3 figures to represent the different patient scenarios, factors that influence the decision-making when managing BMs, and a suggested treatment algorithm.

Reviewer 2 Report

Very interesting review. My only concern is the absence of reference to the type of metastatic disease. I think that a section should be added dealing with 1. oligometastic disease at presentation and results of published studies on loco-regional treatment of primary and brain metastasis; 2. oligoprogressive disease (at CNS) after remission of initial disease. 

Author Response

Response to Reviewer 2

Comment: Very interesting review. My only concern is the absence of reference to the type of metastatic disease. I think that a section should be added dealing with 1. oligometastic disease at presentation and results of published studies on loco-regional treatment of primary and brain metastasis; 2. oligoprogressive disease (at CNS) after remission of initial disease. 

Answer: Thank you for reviewing our work and your valid comment. A new section has been added entitled “Oligometastatic CNS disease and “watch and wait” approach for asymptomatic BMs” on page 8 to more thoroughly describe the different types of metastatic brain disease, including oligometastasis, oligoprogression an oligorecurrence. Please note that we have added a brief paragraph discussing the management of the primary tumor in the oligometastatic state, as the main aim of this review is to focus on novel strategies for the brain metastases. We have identified most common scenarios, which include de novo BMs (and where the discussion about using ablative brain-directed treatments over WBRT has already been mentioned in other sections), as well as the CNS oligoprogression while on active targeted treatment. We have also included 3 figures to represent the different patient scenarios, factors that influence the decision-making when managing BMs, and a suggested treatment algorithm.

Round 2

Reviewer 2 Report

Thanks to the authors for the efforts in adding a paragraph on oligometastatic disease. Indeed, I am not convinced that literature review was exhaustive, and I not agree on conclusions.

Oligometastatic disease may deserve local treatment on both metastasis and primary tumor, with very satisfactory results: for example, Loi et al showed (PMID: 30476480) in a monoinstitutional series of 51 patients affected by NSCLC and synchronous oligometastic disease (mainly cerebral) treated by surgery on primary tumor and surgery or RT on metastasis, a median survival of 42 months and overall survival rates of 62% at 2 years and 34.4% at 5 years. Similar results have been shown by other authors. I think that the impact of local treatments on both primary tumor and metastasis cannot be ignored in both test of the review and management proposed in the algorithm.

Author Response

Comment: Thanks to the authors for the efforts in adding a paragraph on oligometastatic disease. Indeed, I am not convinced that literature review was exhaustive, and I not agree on conclusions.

Oligometastatic disease may deserve local treatment on both metastasis and primary tumor, with very satisfactory results: for example, Loi et al showed (PMID: 30476480) in a monoinstitutional series of 51 patients affected by NSCLC and synchronous oligometastic disease (mainly cerebral) treated by surgery on primary tumor and surgery or RT on metastasis, a median survival of 42 months and overall survival rates of 62% at 2 years and 34.4% at 5 years. Similar results have been shown by other authors. I think that the impact of local treatments on both primary tumor and metastasis cannot be ignored in both test of the review and management proposed in the algorithm.

Answer: The authors want to thank the reviewer for the comments. The previously added section has been modified and the oligometastatic NSCLC treatment to both the primary tumor and metastatic sites has been analyzed and described in more depth (please see page 8, last paragraph). Also, the consideration of primary tumor treatment has been included in the algorithm.

Synchronous oligometastatic NSCLC represents a distinct category of patients that is known to achieve better survival if all the sites of macroscopic disease are treated radically. First prospective data from 39 patients were published in 2012 by De Ruysscher et al., claiming that the identification of this favorable subgroup before therapy was essential[89]. Subsequent study from Massachusetts compared 186 NSCLC patients presenting with OMD versus 539 patients with extensive NSCLC disease (more than 5 distant metastatic lesions)[90]. The aggressive local therapy of primary disease (defined as surgical resection and/or definitive radiation) was associated with prolonged survival (HR, 0.65, P=.043) in the OMD group. Patients with BMs represented 29% of the oligometastatic cohort. Good performance status, nonsquamous histology, and limited nodal disease were predictive factors for improved survival. Many other studies have focused on synchronous brain-only oligometastasis or included mainly brain oligometastatic patients in their analysis, confirming these results[91–93]

As the standard therapeutic approach for oligometastatic NSCLC patients remains undefined, a rigorous discussion on individualized therapeutic management is fundamental. A recent work by an Italian group suggested the urge of adequate selection criteria for this subgroup in order to discuss the potential benefit of systemic and radical loco-regional treatments[94]. This study analyzed a total of 281 synchronous oligometastatic NSCLC patients from different centers, who received radical surgical treatment of the primary tumor with or without neoadjuvant/adjuvant therapy, and radical treatment of all metastatic sites. Brain was the most common site of oligometastasis (50.9%). The 2- and 5-year overall survival was 60.4% and 36.1%, respectively. The authors concluded that young age, single metastasis and pN0 disease were factors associated with better prognosis.

Of note is the first multicenter randomized Phase II trial published by Gomez et al. in 2016, that compared aggressive local therapy to standard-of-care maintenance treatment or observation in oligometastatic NSCLC patients, that did not progress after first line systemic treatment[95]. This study was terminated early due to substantial efficacy improvement in the local consolidative therapy group. Thirteen (26.5%) out of 49 randomized patients had metastasis to the brain. The median progression-free survival in the local consolidative therapy group was 11.9 months versus 3.9 months in the maintenance treatment group (HR 0.35, p=0.0054). While awaiting results of ongoing Phase III trials, the decision on primary tumor management should be made on individual basis, with support of the available evidence. In addition to a widely implemented radical management of BMs, primary local tumor treatment should be strongly considered after identification of this particular subgroup of patients. A multidisciplinary consensus statement on definition and staging of synchronous oligometastatic NSCLC has been formulated and should be used for this purpose[96].
